# Can creative activities and mind-body practices help enhance well-being and mental health awareness? An exploratory qualitative study in UK higher education

**Marianna Cortesi**[ID]*, **Federico Pendenza**[ID], **Elizabeth Haddon**[ID], **Andrea Schiavio**[ID]

School of Arts and Creative Technologies, University of York, York, United Kingdom

* marianna.cortesi@york.ac.uk

## Abstract

Creative arts activities and mind-body practices, such as yoga, have been shown to benefit mental health and well-being. Research in higher education highlights the mental health challenges faced by students and staff in tertiary education; however, most studies on the potential of creative arts and mind-body practices have been conducted in the United States, with limited research investigating their impact in the UK higher education context. This qualitative study seeks to examine students' and staff members' views on extra-curricular creative and mind-body practices provided by one UK university, exploring how engagement in such activities can impact their understanding and awareness of mental health and well-being. In addition, it intends to investigate potential barriers to engagement with such activities. Drawing on questionnaire responses from 25 students and 20 staff members, findings highlight the effectiveness of art-based interventions and mind-body practices in raising awareness and understanding of mental health and well-being, while also having the potential to positively impact individuals' mental health and well-being. Although personal interests and time restrictions may limit engagement, such activities were found to foster community-building, a particularly relevant factor in the post-Covid era, as institutions seek to re-engage students and staff through in-person activities. These findings have therefore important implications for the implementation of similar interventions in higher education and beyond to promote mental health and well-being awareness in diverse communities.

## Introduction

Over the past decades, a wealth of research has consistently highlighted the beneficial impact of creative arts and mind-body practices on mental health and well-being. Interventions such as those based on drawing, writing, crafting, and multi-sensory art that facilitates an experiential exploration of the natural environment, among others,

**Data availability statement:** Data cannot be shared publicly as they contain information that may compromise the anonymity of the research context and participants. The data supporting the findings of this study are available from the corresponding author, Marianna Cortesi, upon request, or from the Ethics committee that approved this research project (act-ethics@york.ac.uk).

**Funding:** This research project was funded by the Mentally Fit York Fund, a University of York funding scheme. No official grant number was provided but the details of the fund are on the following webpage: https://www.york.ac.uk/institute-of-mental-health-research/mentallyfityorkfunding/.

**Competing interests:** The authors have declared that no competing interests exist.

have been shown to contribute to improving individuals' psychological well-being [1]. Similarly, a growing range of research has supported the therapeutic role of creative arts in enhancing individuals' mental health, for example in school settings [2], among underprivileged communities [3], and in care settings [4].

Literature on the benefits of mind-body practices is even more abundant. Yoga-based interventions, for instance, were found beneficial in reducing symptoms of depression and anxiety in the general population [5–7], in helping musicians manage performance anxiety [8], in supporting older people's health-related quality of life and mental health [9] and in enhancing school students' awareness of mental health and well-being [10]. Similar findings arose from studies investigating the effects on well-being of practices such as mindfulness [11–13], and meditation [14,15]. While these practices are generally regarded as complementary to, and not a substitute for, medical interventions, they are seen to offer a number of tangible benefits, such as being cost-effective [16] and being an accessible option for individuals who may not be ready to seek mental health treatments [17].

The higher education environment poses significant challenges to the mental health and well-being of its stakeholders. In the UK, for example, from 2017 to 2023 the proportion of students experiencing mental health issues nearly tripled [18], and university staff, facing growing workloads, have reported high rates of depression and burnout [19]. While the benefits of art-based interventions and mind-body practices in higher education settings have been investigated predominantly in the United States [20–24], they remain relatively underexplored in the UK.

The first UK-based pilot study on the impact of participatory visual arts on students' well-being [25] found that students who participated in a 12-week visual arts course reported improvement in "confidence, motivation and in social relationships". Similar results were reported by Kelly [26], who observed a reduction in the stress levels of several academic staff members in one UK university by using a well-established survey tool, The Warwick-Edinburgh Mental Well-being scale [27], supplemented with additional questions in a yoga-mindfulness based pilot intervention. On a larger scale, Galante and colleagues [28] found that mindfulness could serve as an effective component of a comprehensive mental health strategies for higher education students in the UK.

Despite these insights, research examining the relationship between art-based intervention, mind-body practices and the well-being of students and staff in the UK tertiary-level sector is still in its early stages. This qualitative study contributes to bridging this gap by investigating how engagement in extra-curricular activities such as mind-body practices and artistic activities, offered by one UK university, influences the understanding and awareness of mental health and well-being among the participating students and staff. This is particularly relevant considering that higher education students are at an age that coincides with the peak onset period for mental health issues [29]. In addition to exploring how these activities may develop awareness of mental health and well-being, the present study also focuses on the following aims:

i. to investigate participants' perceptions of the benefits that these activities may provide for their mental health and well-being;

ii. to examine the promotion and facilitation of these activities within the specific educational context;

iii. to highlight the participants' suggestions for further improvements.

By exploring the connection between participation in these activities and the participants' own perception of mental health and well-being, this research aims to provide insights that could inform a strategic implementation of mental health and well-being support for students and staff in UK higher education.

## Materials and methods

### The research context

As will become evident in this and the following section, the design and writing of this study adhere to the Consolidated Criteria for Reporting Qualitative Studies (COREQ) [30], with necessary adaptations for a questionnaire-based study, as COREQ were originally designed for interviews and focus groups. At the time of data collection – July 2023 – the Research & Study Centre (RSC) of one UK university had been offering, since Autumn 2022, five social and creative activities on a regular basis to all Arts and Humanities postgraduate students (taught and research) and staff (academic and administrative). All the activities took place on one of the university campuses and were designed to be optional and flexible, giving attendees complete autonomy over their participation. Similarly, people with all levels of subject knowledge and experience in each of the activities were welcome. Three members of our research team attended some of these activities to gain first-hand experience and insights. The activities were:

• *Yoga*. One-hour morning session on a weekly basis facilitated by an external practitioner. The sessions included some physical exercises and verbal instructions from the facilitator. Some background instrumental relaxing music was played throughout via a speaker. *Yoga* took place in the RSC building and when the researchers joined these sessions, about 9-ten people on average took part in these.

• *Mindfulness*. One-hour afternoon session on a weekly basis facilitated by an external practitioner. Sessions included some gentle physical activity and a body scan meditation activity, introducing a slight variation from standard mindfulness sessions, which typically focus solely on meditative practices [31]. As with *Yoga*, verbal instructions were given by the facilitator, who participated alongside the attendees. *Mindfulness* took place in the RSC building and when the researchers joined these sessions, about three people on average took part in these.

• *The Language Group*. Two-hour sessions on a fortnightly basis facilitated by one staff member. These sessions involved discussions, small and large group activities around languages and cultures across the world with offerings that included Italian, French, Japanese, and Chinese. This activity took place in the RSC building and when the researchers joined about 20 people on average took part in the sessions.

• *Green Fingers*. One-hour session on a monthly basis facilitated by various university staff members. This activity involved growing, nurturing and exchanging knowledge about plants. The sessions took place in different spaces, depending on the type of activity; some were located in the RSC building while others took place in the university's greenhouses. When the researchers took part in these activities, about 15 people on average participated.

• *Cake Friday*. One-hour informal activity on a fortnightly basis where volunteers – staff and students – brought home-baked cakes to share with the RSC community to eat while socializing. These activities were not led by a facilitator and the turnout varied hugely.

Besides such regular activities, at the time of the data collection, the RSC hosted a one-day event named the *Creative Arts Day* open to anyone, which included art-based workshops such as knitting, acrylic painting and cross stitching. The workshops were led by university staff and students who volunteered to facilitate these sessions. These activities

were provided with the specific purpose of rebuilding an in-person community of students and staff members within the RSC community after the Covid-19 pandemic. While these activities were not promoted nor designed to directly enhance the well-being of attendees, the sources reviewed in the previous section on similar practices suggest a potential link between participation and well-being awareness. Therefore, this study aims to explore how participation in these activities may enhance attendees' awareness of mental health and well-being needs, without assuming a predetermined causal relationship.

## Study design

In alignment with our research goal of exploring the impact of these activities on attendees' well-being and mental health awareness, we adopted a qualitative phenomenological design to capture the lived experiences and perspectives of our participants. This approach allowed us to uncover rich, descriptive data that can inform future practices and support strategies. The phenomenological approach was selected as the most appropriate method for exploring attendees' unique experiences, as it prioritizes the individuality of subjective experiences over establishing an objective account of them [32]. Accordingly, this article presents and discusses data gathered from one open-ended questionnaire distributed to all staff and students in the Faculty of Arts and Humanities of one UK university. A questionnaire was selected as the method of data collection because it allowed participants to reflect on their experiences in their own time and provided an inclusive means of engaging individuals who might have been less inclined to take part in in-person interviews. Although we explored existing validated surveys for potential use in our study, the uniqueness of the research context warranted the creation of a new survey. The questions we developed were informed by several factors: a review of several similar studies as discussed in the Introduction, our own participation in the activities (see section below), publicly available information on the RSC website, and promotional content from the RSC's email newsletter.

## The insider researcher

At the time of the data collection, the four members of the research team – two males and two females – were members of the research community within the university. Two were experienced researchers and academic staff members and two were post-doctoral students who had recently completed their PhDs – and, prior to the development of the questionnaire, three of us had attended some of the social and creative activities. Therefore, we had a direct occasional involvement with the research context as well as with the population – students and staff members – that we were studying, determining our position as insider researchers [33]. While our attendance at these activities and interaction with the attendees were not frequent enough to establish a traditional ethnographic approach for this study, the insider position informed our knowledge of the research context. During the data collection, we documented our individual reflections on the activities we participated in through field notes, which were stored in a shared, password-protected Google Drive folder [34]. We did so with the specific intention of reflecting on our experiences as insiders by outlining our observations and perceptions of the activities. Individual reflections were only accessible by the four researchers. The purpose of this note-taking was twofold: while this helped us gain an in-depth understanding of the research context and informed the questionnaire development, it equally supported the reflexive process that is necessary to minimize the impact of the biases arising from our involvement in the research context [35]. In particular, ensuring a high level of reflexivity was crucial for maintaining validity and enhancing the credibility of the study [34].

## Data collection

Prior to distribution of the main questionnaire, a pilot questionnaire was sent to one postgraduate student and one staff member employed by the Research & Study Centre (RSC). An information sheet and consent form were electronically embedded at the start of the questionnaire, informing participants about the study's purpose, aims, and their rights.

Following feedback from the two respondents, amendments to the questionnaire were made to improve clarity and reduce its length. The final version of the questionnaire included six closed questions aimed at gathering respondents' demographic data as well as open-ended questions investigating respondents' understanding of well-being as well as individual experiences of these activities. One RSC administrator emailed the questionnaire to the Faculty of Arts and Humanities staff and student mailing list, and one reminder, spaced within two weeks, was sent out by the same administrator.

Since we aimed to capture the perspectives of all those eligible to attend the activities, we employed purposive sampling to ensure a broad and relevant participant pool: the final anonymized questionnaire was administered via email to all postgraduate students and staff within the Faculty of Arts and Humanities in one UK university in July 2023. In addition, the questionnaire format provided participants with the opportunity to reflect thoughtfully before responding and were considered beneficial for reaching those who might not want to participate in in-person interviews [34]. 25 students and 20 staff members returned the questionnaire. An overview of the demographic data and attendance patterns for participants who returned the questionnaire will be now provided.

## Data analysis

This research involved a qualitative, thematic, inductive analysis of the data. The use of thematic analysis was motivated by the necessity of generating codes in a flexible manner in order to illuminate respondents' lived experiences [36]; at the core of our interest was the respondents' subjective experience of the activities as well as how these experiences related to respondents' understanding of mental health and well-being. The research purpose determined our inductive approach to data analysis; the generation of themes and codes was driven by the responses [37,38].

Our awareness of being insiders in the research context shaped the data analysis process. To balance the benefits with the potential for bias, we employed a collaborative and reflexive approach to enhance trustworthiness. We analyzed questionnaire data following the iterative principles of thematic analysis set out by Braun and Clarke [39]: immersion in the data set, code generation, theme identification, code and theme review and writing. To strengthen the data analysis process and give evidence to the theme and codes generated, quotations from respondents are included in the findings [40]. This process, however, was far from being linear; indeed, after the generation of a first set of codes and themes by the first author, these were reviewed and revised by the other three authors during team meetings. The discussion among the research team members was ongoing until we reached a shared view on the codes and themes generated as well as on connections between themes. This process created opportunities to question assumptions, reduce the risk of bias, and ensure that interpretations were not limited to a single insider perspective but critically examined across the team. Not only did this process enable us to identify new and unexpected themes that could not be anticipated [41]; the collaborative effort produced a process of triangulation that enhanced the trustworthiness of our analysis by capitalizing on the individual perspectives offered by team members [42].

The findings will now be presented in two main sections: first, the results from the student cohort, followed by those from the staff cohort. Although the themes identified through thematic analysis were consistent across both groups, the cohorts are presented separately for clarity and ease of comparison.

## Ethical procedure

This research project has been approved via email by the School of Arts and Creative Technologies Ethics Committee at the University of York, and informed written consent was obtained from all respondents. We exercised care in devising the questionnaire: the questions' phrasing was designed to curtail the risk of respondents revealing any potential identifier, and we only asked questions that were strictly connected with the research purpose. In addition, we granted respondents anonymity through the survey platform Qualtrics, which automatically anonymized responses. An information sheet as well as a written consent were provided at the start of the questionnaire, and respondents indicated their agreement by ticking boxes. In accordance with Thompson and Chambers' [43]

statement that researchers should "reflect on in what way the specific context of their proposed study might create vulnerability", we were aware that answering questions relating these activities to respondents' mental health might have been distressing for some or all respondents; therefore, we included a number of free-access mental health resources at the end of the questionnaire.

## Results

The following subsections present the study's findings organized by respondent groups: students and staff members. The analysis of student responses precedes that of staff members. As the same themes appeared, a mirrored structure is used for both groups, covering the following themes: frequency of activity attendance, effectiveness of the activities, engagement and barriers to participation, conceptualization of well-being, and suggestions for improvement.

### Demographic data and attendance patterns

The demographic data and activity attendance patterns of all respondents are presented below in Tables 1 and 2. Since the results are reported separately for students and staff, the demographic information and attendance patterns are also presented separately for each group.

Respondents' attendance patterns are presented below in Table 2.

**Table 1. Demographic data.**

|  | Students (n = 25) | Staff (n = 20) |
|---|---|---|
| **Gender** | | |
| Female | 18 (72%) | 17 (85%) |
| Male | 6 (24%) | 3 (15%) |
| Non-binary | 1 (4%) | |
| **Age group** | | |
| 18-25 | 10 (40%) | |
| 26-32 | 11 (44%) | 5 (25%) |
| 33-40 | 1 (4%) | 1 (5%) |
| Over 40 | 3 (12%) | 14 (70%) |
| **Nationality** | | |
| UK | 11 (44%) | 13 (65%) |
| European Union | 8 (32%) | 5 (25%) |
| Other countries | 6 (24%) | 2 (10%) |
| **Language** | | |
| English | 13 (52%) | 14 (70%) |
| Non-English | 12 (48%) | 6 (30%) |
| **Role (only staff members)** | | |
| Academic | | 14 (70%) |
| Professional service | | 6 (30%) |

**Table 2. Respondent participation in activities.**

|  | Attended one or more activities | Did not attend any activities | Never heard of these activities | No answer |
|---|---|---|---|---|
| Students | 11 | 10 | 3 | 1 |
| Staff | 8 | 7 | 5 | 0 |

### Students

**Frequency of activity attendance.** Among the 25 students who returned the questionnaire, 11 students [Stu_1; 2; 3; 4; 6; 7; 11; 12; 14; 15; 22; 24] took part in one or more activities provided by the RSC. While all of them detailed which activities they attended, only some of them provided insights into the regularity of their attendance. Table 3 summarizes students' attendance patterns.

**Effectiveness.** Nine of the 11 respondents claimed that their well-being benefited in various ways from attending these activities and one [Stu_22] did not. In relation to the benefits experienced, these activities helped these students get some relaxation (n = 6), establish a routine (n = 3), enhanced their opportunities to socialize (n = 6) and offered them a welcome break from studies (n = 2). Furthermore, two respondents referred to the physical benefits obtained: "*Yoga* has been the most beneficial activity. Getting a weekly routine (despite the early hour!) and doing exercise at the very beginning of my working day has been incredibly important for my mental well-being" [Stu_11]. In relation to relaxation, *Yoga* and *Mindfulness* were regarded by several respondents as particularly important and provided some of them with stress coping strategies that they could use in their daily life:

> Physically, the meditation and breathing techniques I have learnt have helped keep me calm in stressful situations and been useful if I have trouble sleeping. They have taught me to be kind to myself and take things slowly, and I have carried these lessons into my daily life [Stu_1].

One respondent acknowledged similar benefits in relation to the plant-based activity: "Learning how to get 'a green thumb' has been so therapeutic and healthy for my mental well-being that I would love that this activity [...] will continue next year" [Stu_11]. The socialization embedded within some of the activities was also appreciated by several respondents; in particular, the activities "allow me to know new people and engage in social activities" [Stu_2] and made another respondent "feel more welcome [and] included in the dept/faculty" [Stu_12].

Unfortunately, no further insight was provided by the student who did not find the activities beneficial. Nonetheless, this student provided a positive recall of two *Yoga* sessions they attended:

> Our teacher [...] was really good and adapted the level to our abilities. She was happy to meet us where we were physically. Everyone was welcome, and yoga mats were provided as well as tea [Stu_22].

Despite the lack of comments on the reasons why this respondent did not find the other activities beneficial, their positive recall of the *Yoga* sessions highlights the importance of adaptability and inclusivity in such activities, which is explored further in the next section.

**Inclusivity.** Students who took part in one or more activities (n = 11) were asked to comment on the level of inclusivity experienced, as inclusivity is seen as an important dimension of well-being [44]. The majority of respondents (n = 8)

Table 3. Students' attendance patterns.

| Activity | Number of respondents attending the activity | Attendance occurrence |
| --- | --- | --- |
| *Yoga* | n = 8 | Regular n = 6. |
| *Mindfulness* | n = 5 | Regular n = 2; Occasional = 2. |
| *Cake Friday* | n = 9 | Regular n = 2; Occasional n = 5. |
| *The Language Group* | n = 1 | No answer provided. |
| *Green Fingers* | n = 2 | Regular n = 2. |

No answer was offered in relation to the attendance to the art-based one-day activity.

regarded these activities as inclusive. One of them stated: "The sessions have been extremely inclusive, open to all without barriers and limits. For instance, *Yoga* and *Green Fingers* have been amazing activities for both beginners and advanced" [Stu_11]. In this regard, one student appreciated that the *Yoga* and *Mindfulness* facilitator encouraged attendees to "not compare ourselves to others" and added:

> She is also approachable and I feel I could tell her if there was something I was struggling with. At the start of *Mindfulness* classes, she asks how we are feeling and if there is anything in particular we would like to focus on [Stu_1].

The friendly attitude of the facilitators was regarded as a determinant of inclusivity by three other students.

In contrast, some respondents offered suggestions on how to enhance the inclusivity of these activities. The format and location of the activities were considered problematic by three students. One of them would have benefited from a hybrid format: "run online events and make your other events [...] available hybrid as standard. There have been numerous events I would have attended online had there been the option" [Stu_21]. Two others suggested reconsidering the location to make it more accessible for students and staff located in other parts of the city.

**Engagement and barriers to participation.** To deepen our understanding of the attendees' investment into the activities, respondents were asked to state what motivated them to attend. Nine students mentioned socialization: by means of example, one student wanted to "meet like-minded people" [Stu_12] and another one to "feel part of a community that goes beyond the PhD, but gathers human beings before gathering researchers" [Stu_11]. Three students were motivated by the physical and/or mental well-being embedded in the activities – "I want to keep my body healthy" [Stu_3] – whilst three others needed a break from their studies. In addition, two students were interested in learning more.

We then asked respondents to state why they thought these activities were provided by the RSC. Responses were oriented towards two aspects: an equal number of students (n = 12) believed that the purpose of the activities was to either enhance attendees' well-being and/or to foster social interactions. Some of them mentioned both aspects as interconnected: "[The purpose of these activities is] for physical health and mental wellness by connecting diverse people with similar interests" [Stu_17]. Some students (n = 3) thought that these activities were meant to offer a break from academic work: "I guess this is a way to help students and staff of the RSC to share common moments outside academia, which helps improve social boundaries" [Stu_4].

Several students took part in some but not all of the activities offered (n = 11). Others did not take part in any (n = 10); therefore, the questionnaire invited respondents to comment on barriers to attendance. 10 students who had taken part in some of the activities claimed that time constraints and other commitments had prevented them from taking part in the others; one respondent stated that "Timing is not very convenient for some of those activities" [Stu_15]. Four students only attended activities they were interested in: "I am not as interested in *Green Fingers* or in the arts/crafts activities as I am in *Yoga*, *Mindfulness* and *Cake Fridays*. I don't particularly enjoy gardening" [Stu_1]. A few students (n = 3) were concerned about the type of social engagement they would have experienced: "I did not attend [*The Language Group*] events because I was concerned that the conversation during the sessions might have led to PhD-related topics, which I preferred to not talk about while attending a mental well-being activity" [Stu_11]. Lastly, two respondents believed that poor advertising impacted negatively on the limited attendance to some of the activities.

Among those who did not attend any of the activities, six of them mentioned time pressure or geographical distance: one student claimed that "I always had other commitments, sadly, e.g., seminars at uni. But would have loved to participate" [Stu_23], while another claimed to "not liv[e] close enough to campus" [Stu_17]. Two students did not attend because they thought it would not have been beneficial for their well-being. In this regard, one of them commented: "I have social anxiety and thought the benefit of the activity was lower than the benefit of staying home" [Stu_13]. These responses highlight a number of barriers to participation, with the quote from Stu_13 emphasizing the importance of

considering how individuals' differing perceptions of well-being may influence their decision to engage in activities that could contribute to it. This is further explored in the next section.

**Conceptualizations of well-being.** Our insight into the relationship between engagement with social activities and well-being was enriched by students' understanding of well-being. For eight of them, positive well-being was related to functioning in everyday life. One student commented: "[Well-being equated to] feeling relaxed most of the time, not having disturbing and disruptive thoughts […] being able to focus and concentrate on different daily tasks" [Stu_15]. Eight others, instead, understood well-being as a balanced attitude to life or a set of peaceful feelings, for example as "a feeling of balance and calmness" [Stu_17] or "when you feel that your life does not overwhelm your mind and your body" [Stu_24]. Three respondents agreed that taking care of oneself was also an important dimension of well-being.

Some students regarded interpersonal relationships or self-acceptance as key to well-being: for Stu_18, well-being "means being comfortable in your body and in yourself". A radically different interpretation of well-being was portrayed by six students, who viewed it as the absence of suffering. One of them believed that being "resilient to negative inputs" [Stu_13] determined well-being.

In specific connection with the activities, respondents provided insights into the responsibility for creating, maintaining, developing, and supporting well-being; four students claimed that this responsibility is primarily individual. In this regard, one of them commented:

I think as much as the session leader may have a goal of improving mental well-being, the responsibility is always with the individuals first and foremost. We have a duty to one another to be kind and be supportive of others' well-being, but this is meaningless if you don't have a desire to look after yourself [Stu_16].

Differently, 12 respondents believed that the responsibility for attendees' well-being lies between individuals, other participants and the facilitator.

The individual cannot thrive unless the activity takes place in a supporting environment. I think it is up to the individual to check in with themselves and figure out how to get the most out of the activity, but they cannot fulfil their goals if the session leader and other participants are disrespectful and discouraging [Stu_13].

Only one participant regarded the session leaders as mainly responsible for the activity attendees' well-being but, unfortunately, they did not comment further.

**Suggestions for improvement.** Following some critical considerations in relation to limited advertising (n = 2), attendance (n = 2) and inclusivity (n = 2) – as previously discussed in the sections 'Engagement and barriers to participation' and 'Inclusivity' - some respondents offered their suggestions on how to improve the activities' provision and promotion. Five of them would enjoy other types of activities, including physical activities-based classes, social opportunities or activities involving charities. Five other respondents suggested to include more art-based sessions; one of them, highlighted the potential benefits of these sessions for mental well-being:

Drawing or art creative sessions could be beneficial for mental well-being, it would not only permit to express emotion and ideas in a new way, but also inspire our academic research. It would also be a place where you can talk to others, so it could be a nice place to socialize [Stu_4].

Three students believed that the activities themselves did not have a substantial influence on the recreation of a post-pandemic community. One of them claimed:

In my opinion, the activities themselves are incredibly good [...]. However, I guess the main issue […] is the actual lack of a community itself. Since the end of the pandemic, I have not seen more than 20 people working at the RSC on a

daily basis […] whilst before the pandemic researchers flocked to the RSC to work in such a beautiful work and community space […]. Therefore, the [admin team] should work on refreshing the RSC community [Stu_11].

Respondents also considered the organization of the activities. Four of them thought that the activities' attendance rate may benefit from reminders whilst two others believed that a more direct involvement of the students in deciding which activities to offer could be useful: "involve students to attend and organize various activities other [than] the ones proposed [would be useful]" [Stu_2]. Lastly, three students suggested reconsidering the timing and length of the activities to make these more accessible; they would prefer either longer sessions or for activities to be set at a time that would not conflict with typical working hours.

### Staff members

**Frequency of activity attendance.**  Eight of the 20 responding staff members took part in one or more activities provided by the RSC (Sta_1; 2; 9; 12; 13; 14; 17; 20). All of them detailed which activities they attended and all but one staff member provided information about the regularity of their attendance. Table 4 summarizes staff members' attendance patterns.

**Effectiveness.**  Seven staff members claimed to have benefited from doing one or more of these activities. Four of them expressed appreciation for the positive effects these activities had on their health. *Yoga*, for example, was regarded as "great for my mental and physical well-being" [Sta_17]. Similarly, Sta_1 commented: "*Mindfulness* was really good and I felt so much better after the session. It was the first time in ages I could relax and stop the anxiety". Another respondent benefited from learning and engaging with the creativity embedded in the *Creative Arts Day*: "*The Creative Arts Day* was so much fun. I learnt so many things, and I had the chance to be creative, which is really good for my mental health because I am a creative person" [Sta_1]. Three respondents enjoyed socializing with other attendees and one of them valued the natural emergence of socialization during the activity: "The focus was not on socialising, which made it easier to socialize. There was no pressure" [Sta_1]. Similarly, Sta_12 claimed:

I think that simply concentrating on an activity for its own sake – whether that is gardening, doing yoga or even eating cake – is more likely to be conducive to the mental well-being of the participants than stressing possible therapeutic benefits.

One respondent reported no benefit from attending one activity, attributing this perception to physical discomfort: "I did not like [*Mindfulness*] as it was physically uncomfortable and painful rolling on the floor" [Sta_13]. This suggests that aspects such as physical discomfort can act as barriers to participation, potentially shaping perceptions of inclusivity negatively. Further insights on this are explored in the next section.

**Table 4. Staff members' attendance patterns.**

| Activity | Number of respondents attending the activity | Attendance occurrence |
|---|---|---|
| *Yoga* | 3 | Regular n = 2; Occasional n = 1. |
| *Mindfulness* | 4 | Regular n = 1; Occasional = 3. |
| *Cake Friday* | 3 | Occasional n = 2. |
| *The Language Group* | 1 | Occasional n = 1. |
| *Green Fingers* | 3 | Regular n = 2; Occasional n = 1. |
| *Creative Arts Day* | 1 | N/A |

**Inclusivity.** Respondents who took part in the activities (n = 8) were also asked to comment on the perceived inclusivity of these. Six respondents perceived most of the activities they attended as inclusive although one of them explicitly noted that they did not wish to generalize their perception: "I'm not sure about the level of inclusiveness. They felt inclusive to me, but I might not be the best person to judge this" [Sta_1]. This respondent did not elaborate further on why they felt unqualified to make this judgment. One participant related the perceived high level of inclusivity to the wide range of "options given for differing proficiencies within the class" [Sta_9] while another one valued these activities as they "help break the social barriers and brought together different groups together in a shared space" [Sta_13]. Two respondents raised some doubts about the inclusivity of some activities; specifically, Sta_2 reported that "For *Cake Friday* you would need to come with someone or you would be on your own" while another respondent wondered "if the *Yoga* class might be a bit intimidating for complete beginners" [Sta_17]. These reflections reveal that staff participants' perceptions of inclusivity were shaped not only by the structure of the activities but also by individual circumstances and expectations. The next section delves into how personal factors influenced their engagement with these activities.

**Engagement and barriers to participation.** To understand staff members' investment in these activities, we asked those who attended to explain their reasons for participating. For three of them, learning was their aim: Sta_12, for example, attended *Green Fingers* "to get some information about gardening". Two others cited personal interest as the reason for attending the *Creative Arts Day* and *Yoga. Cake Friday* was attended by two respondents for the refreshment provided while two others attended some of the activities to enhance their well-being: "I have experience with yoga and find it energising. This is one of the highlights of my week" [Sta_9]. One respondent evidenced socialization as a reason for attendance.

To gather further insights into staff members' willingness to engage with the activities, participants were asked to comment on what they perceived as being the aim of these activities. Seven respondents believed that these activities were provided to enhance the well-being of attendees; however, two of them interpreted this aim as focusing on student welfare rather than that of the staff, claiming that these activities "help create a positive space for students and support their mental wellbeing" [Sta_7]. Five respondents claimed that the activities were aimed at enhancing the sense of community by "shar[ing] resources and support[ing] one another" [Sta_16]. Once again, one respondent suggested that these activities "Help students make connections" [Sta_3], directing their response toward student benefits. Three respondents felt the activities were designed to provide attendees with a break from work while three others considered provision from an institutional perspective; one felt that these activities were provided "because all faculties need to complete inclusion, equality, mental health tick-box exercises" [Sta_15].

Given that none of the staff respondents attended all the activities, all of them (n = 20) were asked to comment on what prevented them from taking part. 12 of them could not attend due to time constraints: "My workload is too heavy. If I'm to get through my work and also manage life at home I cannot take time in the working day for such things" [Sta_10]. Five staff members stated a lack of personal interest: "Yoga is not something that I know I would enjoy personally and physically" [Sta_13]. Lastly, three respondents did not attend these activities due to the fear of being alone, while another held an opposite view, aligning their lack of attendance to the social aspect of the activities: "I would rather have resources for things I could do by myself, at home" [Sta_4]. Such contrasting individual responses likely suggest that factors like socialization, among others, may influence individuals' understanding of well-being in varying ways. Further insights into these dynamics will now be provided.

**Conceptualization of well-being.** Staff members' understanding of well-being provided insights into the relationship between engagement with the activities and well-being enhancement. 14 respondents believed that well-being allowed them to cope with everyday life's demands; one of them described it as "Feeling mentally balanced enough to tackle each day's challenges" [Sta_17]. Six respondents conceptualized well-being as connected with feelings of happiness, enjoyment and/or relaxation: "My definition of mental well-being would be being comfortable, safe, healthy and happy" [Sta_20]. In addition, some respondents had contrasting views on the extent to which separating problems from one's self

equates to positive well-being: while three respondents believed that "to separate from our problems in order to handle them […] is almost like saying we are denying their personal existence or facing them" [Sta_13], two others believed that "the ability to hold events, situations and people as separate from self is an important aspect of well-being" [Sta_10].

Staff respondents were also invited to comment on who they deemed as responsible for the well-being of participants in the context of the activities. Seven respondents stated that the responsibility of participants' well-being is collective:

> Any activities to promote mental health need to be carefully designed to ensure they are supportive of the individual so I would suggest that the session leader has responsibility to ensure proper session design and participants have a responsibility to respect the needs of other participants [Sta_19].

Differently, four respondents acknowledged an individual responsibility for well-being, with one of them claiming: "It is the individual's responsibility to engage in activities enough for mental well-being benefits [Sta_19]. Lastly, three respondents focused on the role of the employer in ensuring individuals' well-being in a workplace: "In a work environment the employer has a very large responsibility for the mental well-being of their staff (and in this case also students)" [Sta_7]. Participants also shared insights into how these activities might be enhanced to benefit attendees, offering suggestions for improvement that will now be explored.

**Suggestions for improvement.** Staff members provided some suggestions on how to improve the social and creative activities provision. Some of them would value a broader range of activities, such as social activities (n = 2), creative activities (n = 2), music-based activities (n = 2) and walks (n = 3), with "campus walks [being regarded as] a good opportunity to do something active but also talk to others" [Sta_3]. Two respondents suggested modifying the timing of the sessions by alternating session days each week: "the time could be alternated from week to week so that those who work part time have more chance to attend" [Sta_2], and one respondent would like to have longer sessions – half a day – for creative activities.

Four respondents expressed criticism towards the university, advocating for a different approach towards promoting well-being. While two of them believed that resources such as "better car parking provision for staff, bike sheds, cheaper bus fares and having refreshments provided at events" [Sta_15] would be more beneficial than the activities, Sta_7 claimed:

> I'm not necessarily sure that adding more activities is the way to go. The whole thing needs to be turned on its head and approached from a different angle. How can the roots and causes of bad mental well-being […] across our faculty be tackled? What can be done about workload, pay and other difficult working conditions? Yes of course this takes more effort than putting on a few yoga classes, but if the university is actually serious about mental well-being then this is the approach it should be taking.

These suggestions highlight a range of perspectives on how the activities could be enhanced, while also raising broader concerns about institutional responsibility in addressing well-being beyond offering extracurricular activities.

## Discussion

This research explored how awareness of well-being and mental health among students and staff may be shaped by social and creative activities at one UK university. Findings from this UK-based study generally align with research from the United States on the impact of art-based interventions and mind-body practices, where evidence suggests these interventions can positively affect participants. However, several US studies were conducted as controlled, designed interventions [20,22,24], often demonstrating short-term benefits that may have been influenced by the limited duration of the projects [24]. In contrast, our research explored respondents' perceptions of their own well-being in relation to activities that were already in place prior to the study, providing insights into the effects of ongoing pre-existing activities rather than

structured or quasi-experimental interventions. All respondents were asked to describe their personal conceptualization of well-being. Most of them equated well-being with functioning effectively in everyday life, while some students, but not staff members, also identified it as the absence of pain. Although optimal well-being and mental health are now recognized as not merely the absence of disease but relate to individuals' ability to "cope with the normal stresses of life" [45], these different conceptualizations provided insightful information on respondents' expectations of the activities. For example, the vast majority of respondents acknowledged well-being in the context of the activities offered as a collective responsibility shared between the facilitators and the participants. This suggests that the success of such activities in relation to participants' well-being depends not only on the activities themselves but also on the environment created by the facilitators and the collaborative atmosphere among participants.

Most respondents who participated in the activities considered them effective, and several, especially those involved in *Yoga* and *Mindfulness*, regarded them as relaxing, which may relate to the high stress and workload experienced in UK higher education [46] and decreased levels of optimal mental health [18,47]. Socialization was another positive effect noted by participants. Students were more likely to identify socialization as an explicit benefit, while at least one staff member, Sta_1, valued the implicit socialization inherent in the activities, which was not overtly promoted. These implicit benefits align with the notion of hidden curriculum, intended as "the often unspoken and sometimes unintended outcomes of participating in the educational system" [48]. This suggests that how institutions present and promote well-being activities can influence participation, and offering opportunities for socialization without explicit emphasis may allow individuals to benefit from it by acknowledging that socialization is experienced differently by each participant [49].

Most respondents commented positively on the inclusivity of the activities, with some appreciating the opportunity to meet people from various backgrounds – an increasingly relevant aspect in UK higher education, given the number of international students [50]. Inclusivity was also noted in relation to mind-body practices being accessible to individuals of any experience level, supporting wider participation. Surprisingly, none of the participants commented on aspects such as physical spaces and use of language by the facilitators, which have been identified as determinants of inclusivity in *Yoga* [51]. However, one staff member regarded *Mindfulness* as physically painful, raising questions about its accessibility. While mindfulness is considered a "universal human capacity" [52] focused on mental awareness and, unlike yoga, typically not involving physical exercise [31], the sessions offered by the RSC included some gentle physical activity, which may have legitimately contrasted with some participants' expectations. This highlights the importance of transparency in managing expectations and accommodating diverse abilities and preferences.

Another theme emerging from data analysis was participants' engagement with these activities. Both staff and students acknowledged well-being as a key motivator for attendance, supporting our suggestion of a potential connection between participation and enhanced well-being. While at the time of data collection participants' engagement with the activities had been relatively brief – less than one year for early starters – these findings have relevant implications for higher education institutions; while prior research indicates that interventions are often explicitly aimed at well-being enhancement [53,54], this study indicates that social and creative activities can provide well-being benefits even when their primary goal – in this case, rebuilding a post-Covid-19 community – does not explicitly target it. Motivations to participate differed between students and staff: many students were drawn by the prospect of socializing, in alignment with the role of socialization in students' social and academic integration [55,56], whereas staff were primarily motivated by personal interest or a desire to learn. While it seems that staff were less interested in socialization – contradicting evidence that reduced social interaction during the Covid-19 pandemic adversely affected staff mental health [57] – responses from two staff members, Sta_3 and Sta_7, suggested that they viewed socialization primarily as a key benefit for students. This seems to indicate that they perceived the activities as designed for students' needs, rather than their own. Again, transparency about the objectives and the intended recipients of these types of activities may enhance the capacity of all participants – both students and staff – to make informed decisions regarding their attendance.

Respondents identified barriers to attendance, with staff and students agreeing that lack of personal interest and time constraints prevented them from participating. Time limitations are unsurprising, as similar challenges have been observed in studies on mental health-based interventions among doctors [58] and students [59]. In addition, it could be reasonable to suppose that the heavy workload reported by staff could have contributed to such perception. To support a post-pandemic community, respondents suggested incorporating a broader range of activities including hybrid format. Although digital delivery may seem in contrast with the goal of rebuilding in-person communities, digital technologies can play a key role in "supporting new places and community building processes". [60]. Nonetheless, some students felt the university's efforts to recreate a post-pandemic community were insufficient, and the extended use of online tools during the Covid-19 pandemic may have exacerbated isolation [61], even as digital flexibility helped others adjust to in-person education [62]. Thus, a hybrid model may meet diverse needs, but incentivizing in-person participation remains crucial to fostering a vibrant post-pandemic community.

## Limitations

While the insights provided by respondents were highly relevant, a few limitations should be acknowledged. The study was conducted on a small-scale basis, with participants from a single institution, which may limit the generalizability of the findings. Moreover, although the questionnaire offered valuable insights from 45 respondents, its format may have constrained the depth of responses. Future research involving larger samples, multiple higher education institutions, and mixed-methods studies would help to build on and extend the findings of this study. In addition, while self-reported data offer valuable insights into participants' experiences, they are inherently subject to potential inaccuracies, such as social desirability bias or unintended memory lapses. Another limitation relates to potential self-selection bias, since participants of the activities chose whether to take part in the research project or not. Consequently, the data may not fully represent the broader population of students and staff involved in the activities but rather reflect the perspectives of those who opted to participate in the research. This approach was however deemed necessary to ensure ethically sound research, where participation is entirely voluntary. The authors remained mindful of this limitation throughout all stages of the study, from design to analysis and publication.

## Conclusions

The benefits of art-based interventions and mind-body practices in UK higher education remain largely underexplored. The findings of this study address this gap by suggesting a generally positive reception of the RSC activities from both students and staff. This is particularly relevant when considering the period after the Covid-19 lockdowns, where universities needed to navigate the transition back to in-person education by re-establishing a sense of community among students [63]. Beyond the specific context of this research, similar activities could be implemented across other higher education institutions or community-building programs to encourage engagement, social connections, and support mental health.

Another consideration emerging from these findings is the positive impact these activities appear to have had on attendees' well-being. Although the activities were not explicitly designed to promote well-being, several participants reported experiencing well-being benefits. For higher education institutions facing growing concerns around students and staff mental health, this suggests that offering activities that are not directly marketed as well-being interventions but inherently incorporate elements of well-being could be a valuable strategy for institutions. Nonetheless, the increasing demands in higher education [64] indicate that institutional endorsement of such activities is often at odds with staff workload expectations. This necessitates that higher education institutions reflect on how to effectively negotiate the promotion of these activities with the workload they require from their employees.

Beyond the higher education sector, similar activities might be considered to raise mental health and well-being awareness in other settings such as schools and community centres. While it is acknowledged that financial and logistic

constraints might pose barriers, our findings suggest that offering these, or similar activities might be a springboard for community building among attendees. It is nonetheless important to emphasize that these activities should be viewed as supplementary opportunities that can contribute to enhancing participants' well-being alongside professional mental health support structures.

## Acknowledgments

The authors would like to thank all the participants for their time and involvement.

## Author contributions

**Conceptualization:** Marianna Cortesi, Federico Pendenza, Elizabeth Haddon, Andrea Schiavio.

**Data curation:** Marianna Cortesi.

**Formal analysis:** Marianna Cortesi.

**Funding acquisition:** Elizabeth Haddon.

**Investigation:** Marianna Cortesi, Federico Pendenza.

**Methodology:** Marianna Cortesi, Federico Pendenza.

**Project administration:** Elizabeth Haddon.

**Supervision:** Elizabeth Haddon, Andrea Schiavio.

**Writing – original draft:** Marianna Cortesi.

**Writing – review & editing:** Marianna Cortesi, Federico Pendenza, Elizabeth Haddon, Andrea Schiavio.

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
