## [Decision Letter · Decision Letter 0]

7 Aug 2025

PONE-D-25-29029Can creative activities and mind-body practices help enhance well-being and mental health awareness? An exploratory qualitative study in UK higher educationPLOS ONE?

Dear Dr. Cortesi,

Thank you for submitting your manuscript to PLOS ONE. After careful consideration, we feel that it has merit but does not fully meet PLOS ONE’s publication criteria as it currently stands. Therefore, we invite you to submit a revised version of the manuscript that addresses the points raised during the review process.

We look forward to receiving your revised manuscript.

Kind regards,

Jenna Scaramanga

Staff Editor

PLOS ONE

Journal Requirements:

 “This research project was funded by the Mentally Fit York Fund, a University of York funding scheme. No official grant number was provided but the details of the fund are on the following webpage: https://www.york.ac.uk/institute-of-mental-health-research/mentallyfityorkfunding/”

4. In this instance it seems there may be acceptable restrictions in place that prevent the public sharing of your minimal data. However, in line with our goal of ensuring long-term data availability to all interested researchers, PLOS’ Data Policy states that authors cannot be the sole named individuals responsible for ensuring data access (http://journals.plos.org/plosone/s/data-availability#loc-acceptable-data-sharing-methods).

Additional Editor Comments:

The reviewer has raised a number of concerns that need attention. 

 Please note that we have only been able to secure a single reviewer to assess your manuscript. We are issuing a decision on your manuscript at this point to prevent further delays in the evaluation of your manuscript. Please be aware that the editor who handles your revised manuscript might find it necessary to invite additional reviewers to assess this work once the revised manuscript is submitted. However, we will aim to proceed on the basis of this single review if possible. 

Reviewers' comments:

Reviewer's Responses to Questions

**Comments to the Author**

1. Is the manuscript technically sound, and do the data support the conclusions?

Reviewer #1: Yes

2. Has the statistical analysis been performed appropriately and rigorously?

Reviewer #1: N/A

3. Have the authors made all data underlying the findings in their manuscript fully available?

Reviewer #1: No

4. Is the manuscript presented in an intelligible fashion and written in standard English?

Reviewer #1: Yes

Reviewer #1: This is an engaging and timely manuscript that addresses an important gap in the literature on creative and mind-body practices within the UK higher education context. The study offers valuable exploratory insights into how such activities may foster well-being awareness, social connection, and community building among students and staff. The use of an open-ended questionnaire and thematic analysis is appropriate, and the findings are presented with clarity, supported by rich participant quotes. The manuscript is sound, well-written, and meets PLOS ONE’s publication criteria. With minor revisions—mainly clarifying data availability, expanding the discussion of limitations, and tightening the narrative—it will be a strong contribution to the literature.

Strengths:

Clear rationale for the study situated within relevant literature.

Appropriate qualitative design with transparent analytic procedures: The study is methodologically appropriate for an exploratory qualitative design. The use of an open-ended questionnaire analyzed via thematic analysis is consistent with the study’s aims. The conclusions drawn—mainly that creative and mind-body activities may enhance well-being awareness and community building—are supported by the data presented. However, the authors should better clarify the limitations of self-reported data, the relatively small sample size, and the absence of triangulation beyond researcher review.

Inclusion of both students and staff offers a broader institutional perspective.

Practical implications for higher education are well articulated.

Suggestions for improvement:

Clarify limitations. Explicitly acknowledge the limited sample size and self-selection bias, and explain how these may affect the transferability of findings.

Refine data availability statement. Provide a more detailed description of how interested researchers can access de-identified data or analytic materials while maintaining participant anonymity. The data availability statement indicates that data cannot be shared publicly due to confidentiality concerns but may be available on request from the corresponding author. While this is permissible under PLOS ONE policy in specific cases (e.g., to protect participant anonymity), the authors should provide greater detail on what data will be shared and the process for access (e.g., de-identified excerpts, coding framework). This would improve transparency.

Streamline sections of the manuscript. Consider condensing the literature review and parts of the discussion where points are repeated, focusing on the most salient findings and implications.

Broaden the implications. Consider including a brief reflection on how these findings might inform institutional policy beyond the immediate context (e.g., implications for other universities or community-based programs).

Address researcher positionality. While insider status is acknowledged, more discussion of how this was managed during data collection and analysis (e.g., reflexive journaling or audit trails) would strengthen trustworthiness.

**Do you want your identity to be public for this peer review?** For information about this choice, including consent withdrawal, please see our Privacy Policy

Reviewer #1: **Yes: ** Tonje M. Molyneux, PhD

---

## [Author Response · Author response to Decision Letter 1]

17 Sep 2025

Please find below my responses to the Reviewer, as also outlined in the 'Response to reviewers' letter.

1) Comment: Clarify limitations. Explicitly acknowledge the limited sample size and self-selection bias, and explain how these may affect the transferability of findings.

Response: A dedicated section on limitations has been added following the ‘Discussion’. This section explicitly acknowledges self-selection bias, potential inaccuracies inherent in self-reported data, the small sample size, and its implications for the generalizability of the findings.

2) Comment: Refine data availability statement.

Response: The ‘Data availability statement’ has been updated in alignment with the reviewer (and the journal)’s comment.

3) Comment: Streamline sections of the manuscript. Consider condensing the literature review and parts of the discussion where points are repeated, focusing on the most salient findings and implications.

Response: Sections of the ‘Discussion’ have been streamlined, reducing the word count from approximately 1300 to 900 words, and the discussion of findings has been presented in a more integrated manner to enhance readability and impact. The Literature review (Introduction) has not been further shortened (remaining around 500 words), as it was already heavily streamlined during manuscript drafting. The authors believe that additional cuts could compromise the clarity of the research rationale and weaken the demonstration of a thorough review of relevant literature.

4) Comment: Broaden the implications. Consider including a brief reflection on how these findings might inform institutional policy beyond the immediate context (e.g., implications for other universities or community-based programs).

Response: The Conclusions section has been revised to broaden the implications of our findings, highlighting their relevance for other higher education institutions and community-based programs.

5) Comment: Address researcher positionality. While insider status is acknowledged, more discussion of how this was managed during data collection and analysis (e.g., reflexive journaling or audit trails) would strengthen trustworthiness.

Additional details have been added to both the ‘The Insider Researcher’ and ‘Data Analysis’ sections to clarify how we managed our positions as insider researchers and ensured the trustworthiness of the findings.

---

## [Decision Letter · Decision Letter 1]

16 Oct 2025

PONE-D-25-29029R1Can creative activities and mind-body practices help enhance well-being and mental health awareness? An exploratory qualitative study in UK higher educationPLOS ONE?

Dear Dr. Cortesi,

Thank you for submitting your manuscript to PLOS ONE. After careful consideration, we feel that it has merit but does not fully meet PLOS ONE’s publication criteria as it currently stands. Therefore, we invite you to submit a revised version of the manuscript that addresses the points raised during the review process.

We look forward to receiving your revised manuscript.

Kind regards,

Hariom Kumar Solanki, M.D.

Academic Editor

PLOS ONE

Journal Requirements:

Reviewers' comments:

Reviewer's Responses to Questions

**Comments to the Author**

Reviewer #1: All comments have been addressed

Reviewer #2: (No Response)

Reviewer #3: All comments have been addressed

2. Is the manuscript technically sound, and do the data support the conclusions?

Reviewer #1: Yes

Reviewer #2: Partly

Reviewer #3: Yes

3. Has the statistical analysis been performed appropriately and rigorously?

Reviewer #1: N/A

Reviewer #2: N/A

Reviewer #3: Yes

4. Have the authors made all data underlying the findings in their manuscript fully available?

Reviewer #1: Yes

Reviewer #2: Yes

Reviewer #3: Yes

5. Is the manuscript presented in an intelligible fashion and written in standard English?

Reviewer #1: Yes

Reviewer #2: Yes

Reviewer #3: Yes

Reviewer #1: Thank you for your thorough revisions. I have reviewed the revised manuscript and am satisfied that you have adequately addressed all of my concerns.

The new limitations section appropriately acknowledges sample size, self-selection bias, and self-reported data constraints. The data availability statement now includes proper institutional contact information. The Discussion section is more concise and focused with improved flow, and I accept your rationale for maintaining the Introduction's current length. The Conclusions section effectively situates your findings for broader application beyond the immediate study context. The enhanced descriptions of reflexive practices and collaborative analysis procedures substantially strengthen the methodological rigor and trustworthiness of the study. These revisions have improved the manuscript's overall clarity and scholarly contribution.

Reviewer #2: - Expand the limitations section to discuss how questionnaire format may limit depth and suggest future mixed-methods follow-ups.

- Add a brief subsection in discussion on how findings align with or diverge from US studies cited, to strengthen the UK gap argument.

- Consider including a supplementary file with the full questionnaire for transparency.

Reviewer #3: The Author is able to respond to the initial comments made. However, there are gaps that need to be covered including:

1. Under methodology, the study design is not made clear, this needes to be stated clearly.

2. Under results section, last column of Table 2 is not clear; it needs to be made clear.

If these can be addressed, the article can be published, it is good based on the subject matter and the limitations highlighted regarding the study.

**Do you want your identity to be public for this peer review?** For information about this choice, including consent withdrawal, please see our Privacy Policy

Reviewer #1: No

Reviewer #2: **Yes: ** Dr Syed Irfan Ali

Reviewer #3: **Yes: ** Abdulrahman Ahmad

---

## [Author Response · Author response to Decision Letter 2]

5 Nov 2025

The following summarizes the revisions made in response to the reviewers’ comments.

Reviewer 2 comments

1) Comment: Expand the limitations section to discuss how questionnaire format may limit depth and suggest future mixed-methods follow-ups.

Response: The limitations of the questionnaire format have been addressed in the ‘Limitations’ section, explicitly noting the potential constraints on response depth and recommending that future research adopt a mixed-methods approach.

2) Comment: Add a brief subsection in discussion on how findings align with or diverge from US studies cited, to strengthen the UK gap argument.

Response: At the beginning of the Discussion section, a brief summary has been added to highlight how the findings of this study both align with and differ from US-based research. In particular, the shared positive effects of creative activities and mind-body practices are acknowledged, while the distinction between the controlled interventions typical of some US studies and the investigation of pre-existing activities in this UK-based study is clearly noted.

3) Comment: Consider including a supplementary file with the full questionnaire for transparency.

Response: Unfortunately it is not possible to include the full questionnaire as a supplementary file, as it contains identifying references that could, directly or indirectly, compromise the privacy of individuals involved in the activities. However, as added in the Data Availability statement, a copy of the questionnaire can be provided upon reasonable request.

Reviewer 3 comments.

1) Comment: Under methodology, the study design is not made clear, this needs to be stated clearly.

Response: In the ‘Study Design’ section we added a statement that clarifies why the questionnaire was chosen as the data collection tool as opposed to others such as interviews. The ‘Materials and Methods’ section, along with its subsequent subsections, addresses all key components of the design, including the research aim and rationale, the methodological approach (qualitative phenomenological design), data collection, participant selection, and data analysis. The authors, supported by feedback from the first reviewer, consider the study design to be clear and consistent with reporting practices in recent qualitative publications in PLOS ONE (e.g. Khan et al., 2025).

2) Comment: Under results section, last column of Table 2 is not clear; it needs to be made clear.

Response: The last column of Table 2 has been revised for clarity to indicate that participants did not respond to the question addressed in the Table. In the staff responses, ‘0’ has been used instead of ‘N/A’ to avoid confusion.

---

## [Editor Report · Decision Letter 2]

30 Nov 2025

Can creative activities and mind-body practices help enhance well-being and mental health awareness? An exploratory qualitative study in UK higher education

PONE-D-25-29029R2

Dear Dr. Cortesi,

We’re pleased to inform you that your manuscript has been judged scientifically suitable for publication and will be formally accepted for publication once it meets all outstanding technical requirements.

Kind regards,

Hariom Kumar Solanki, M.D.

Academic Editor

PLOS ONE
---

## [Editor Report · Acceptance letter]

PONE-D-25-29029R2

PLOS One

Dear Dr. Cortesi,

I'm pleased to inform you that your manuscript has been deemed suitable for publication in PLOS One. Congratulations! Your manuscript is now being handed over to our production team.

Kind regards,

on behalf of

Dr. Hariom Kumar Solanki

Academic Editor

PLOS One